# Label-Free Raman Microspectroscopy for Identifying Prokaryotic Virocells

Indra Monsees,[a] Victoria Turzynski,[a] Sarah P. Esser,[a] André Soares,[a] Lara I. Timmermann,[a] Katrin Weidenbach,[b] Jarno Banas,[e] Michael Kloster,[c] Bánk Beszteri,[c] Ruth A. Schmitz,[b] Alexander J. Probst[a,d]

aGroup for Aquatic Microbial Ecology, Environmental Microbiology and Biotechnology, University Duisburg-Essen, Essen, Germany
bInstitute for General Microbiology, Christian Albrechts University, Kiel, Germany
cPhycology Group, Faculty of Biology, University Duisburg-Essen, Essen, Germany
dCentre of Water and Environmental Research (ZWU), University of Duisburg-Essen, Essen, Germany
eEssen, Germany

**ABSTRACT** Raman microspectroscopy has been used to thoroughly assess growth dynamics and heterogeneity of prokaryotic cells, yet little is known about how the chemistry of individual cells changes during infection with virulent viruses, resulting in so-called virocells. Here, we investigate biochemical changes of bacterial and archaeal cells of three different species in laboratory cultures before and after addition of their respective viruses using single-cell Raman microspectroscopy. By applying multivariate statistics, we identified significant differences in the spectra of single cells with/without addition of virulent dsRNA phage (*phi6*) for *Pseudomonas syringae*. A general ratio of wavenumbers that contributed the greatest differences in the recorded spectra was defined as an indicator for virocells. Based on reference spectra, this difference is likely attributable to an increase in nucleic acid versus protein ratio of virocells. This method also proved successful for identification of *Bacillus subtilis* cells infected with the double-stranded DNA (dsDNA) phage *phi29*, displaying a decrease in respective ratio, but failed for archaeal virocells (*Methanosarcina mazei* with the dsDNA methanosarcina spherical virus) due to autofluorescence. Multivariate and univariate analyses suggest that Raman spectral data of infected cells can also be used to explore the complex biology behind viral infections of bacteria. Using this method, we confirmed the previously described two-stage infection of *P. syringae*'s *phi6* and that infection of *B. subtilis* with *phi29* results in a stress response within single cells. We conclude that Raman microspectroscopy is a promising tool for chemical identification of Gram-positive and Gram-negative virocells undergoing infection with virulent DNA or RNA viruses.

**IMPORTANCE** Viruses are highly diverse biological entities shaping many ecosystems across Earth. However, understanding the infection of individual microbial cells and the related biochemical changes remains limited. Using Raman microspectroscopy in conjunction with univariate and multivariate statistics, we established a marker for identification of infected Gram-positive and Gram-negative bacteria. This nondestructive, label-free analytical method at single-cell resolution paves the way for future studies geared towards analyzing virus-host systems of prokaryotes to further understand the complex chemistry and function of virocells.

**KEYWORDS** bacteriophage, phage, phi29, phi6, virus

Viruses substantially influence global ecosystems and biogeochemical cycles by infecting host populations. Predation can cause release of organic carbon and also enhance horizontal gene transfer (1), as viruses can act as mobile genetic elements (MGEs). Those viruses that infect bacteria have the specific denotation bacteriophages, or just phages (2). Viruses are generally differentiated based on the type of genetic information stored in their viral

Address correspondence to Indra Monsees, Indra.monsees@uni-due.de, or Alexander J. Probst, alexander.probst@uni-due.de.

The authors declare no conflict of interest.

[This article was published on 15 February 2022 with an error in Acknowledgments. A name was removed in the current version, posted on 24 February 2022.]

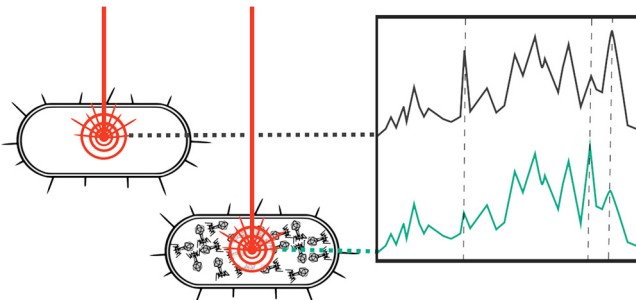

**FIG 1** Conceptual figure for the study. Laser of the Raman microspectroscope is focused on a single microbial cell. The presence of virions replicated inside the cell alters the Raman spectrum, especially in the determined areas. Virocells can be determined by calculation of a ratio based on these intensities.

particle, either single- or double-stranded DNA (dsDNA) or RNA (3). Prokaryotic viruses are also categorized based on their reproduction cycle as lysogenic or lytic (although other strategies, like chronic infection or pseudolysogeny, have been reported infrequently [4]). Viruses can insert their genome into the plasmid or genome of an infected host and proliferate along with host reproduction (lysogeny). A lytic strategy involves the reorganization of host metabolism envisaging reproduction of virions and, ultimately, cell lysis. A host cell infected with a virulent virus is referred to as a virocell and needs to be differentiated from ribocells, cells that generally proliferate irrespective of an infection (5). In a recent study, transcriptomics and proteomics were used to investigate whether metabolic differences between uninfected cells and virocells can impact an entire ecosystem (6). However, the study of virocells generally necessitates nondestructive techniques that can capture virocell characteristics at the single-cell level prior to cell lysis.

The development of confocal Raman microspectroscopy has enabled the measurement of single microbial cells (7), which consequently opened the possibility to gain insights into the heterogeneity of microbial communities (8). The combination of Raman microspectroscopy instruments with multivariate data analysis of digitally recorded spectra allowed for further increases in sensitivity in the last 2 decades, resulting in the detection of biochemical differences between bacterial species across growth phases (9). In this context, multivariate statistical analysis of Raman spectra has been used to differentiate single cells based on discrete wavenumbers corresponding to biochemical compounds. Huang and colleagues described a correlation between the fraction of $^{13}$C in the carbon source and a ratio shift based on Raman peaks of unlabeled [$^{12}$C] phenylalanine and $^{13}$C-labeled phenylalanine (10). The ratio between isotopically labeled and unlabeled molecules can be applied to identify key degraders in mixed cultures and allows specific cell sorting for single-cell methods (11). However, this sensitivity is also the bottleneck of this technique, as demonstrated by García-Timermans et al., who highlighted the influence of the sample preparation on the recorded spectra (12). Such differences complicate the construction of a public database and comparability of spectra across studies. Nevertheless, the comparison of selected wavenumbers between individual spectra of a single study is crucial for expeditious categorization of single cells based on their chemical composition.

In this study, the high sensitivity of Raman microspectroscopy to identify and characterize microstructural intracellular changes as well as viruses and their effects on host metabolism was used to test the suitability of this technology for differentiating uninfected cells from virocells (Fig. 1). To this end, three different model host-virus systems, including virulent DNA and RNA viruses, were used to analyze and monitor chemical changes during infection at the single-cell level using Raman microspectroscopy.

## RESULTS

**Significant differences in the chemical composition of infected and uninfected cultures of *P. syringae*.** For establishing the differences between virocells and noninfected cells, we used the well-known virus host system of *P. syringae* (Gram-negative)

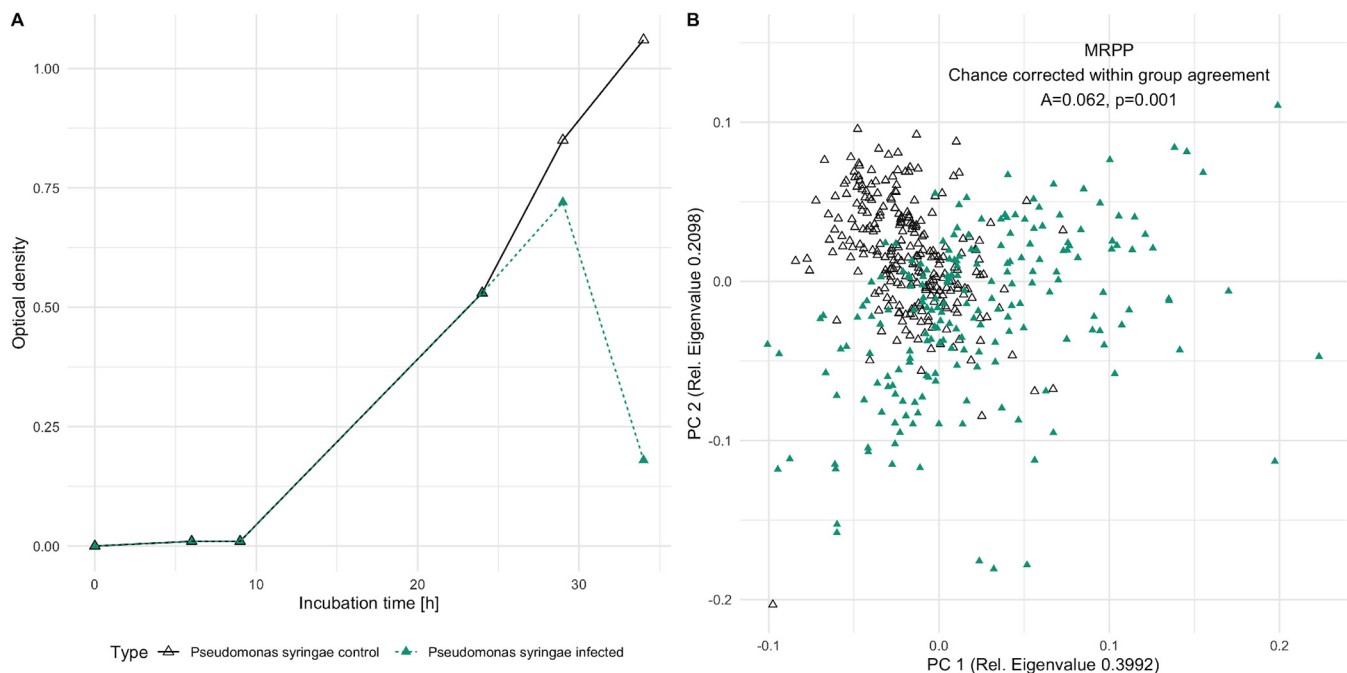

**FIG 2** *Pseudomonas syringae* cultures with (green, filled triangles) and without (black, empty triangles) addition of phage *phi6* after 24 h. (A) Growth curve determined by optical density. The drop in optical density corresponds to phage-mediated cell lysis after 34 h. (B) Principal component analyses of *P. syringae* single-cell Raman spectra after lysis (after 34 h) (ordination analyses based on Euclidean distance and spectral contrast angle revealed nearly identical results; see Fig. S1) and the result of the MRPP for control sample versus infected sample.

and its phage, *phi6* (13). *phi6* has already been studied via Raman microspectroscopy in past decades but never in association with its host (14). *phi6* is a double-stranded RNA (dsRNA) phage of the order *Mindivirales* (15, 16), and its maturation is described to take place in two steps (17). Addition of *phi6* to *P. syringae* cultures resulted in the expected decline in optical density, enabling us to harvest cell populations representing a mixture of virocells and uninfected cells (Fig. 2A). We used this cell population and a culture without phage addition for comparison in single-cell Raman microspectroscopy. In doing so, we successfully measured 448 high-quality spectra of individual *P. syringae* cells, of which 198 cells were measured after addition of *phi6*. The other 250 spectra were reference spectra from uninfected cells of *P. syringae*. Inspection of the spectra and comparison to previously published Raman spectra of bacteria confirmed the typically expected peaks for biomolecules, confirming the measurement of actual microbial cells (9).

Using the individual spectra of each measured cell, we computed an ordination analysis comparing individual cells of cultures with and without phage addition, which showed substantial differences (Fig. 2B). Importantly, the two data sets (with and without phage addition) were not entirely separated along principal component 1 (PC 1) or PC 2 but showed differences along both PCs, which agrees with the above-mentioned mixture of virocells and uninfected cells in populations after phage addition. To test the observed differences for significance, a multiresponse permutation procedure (MRPP) was applied, because in addition to the *P* value for significance, it provides chance-corrected within-group agreement (A), which displays the difference between the groups (18, 19). MRPP analysis displayed a highly significant *P* value (<0.001), with a chance-corrected within group agreement of 0.062. Consequently, phage addition and infection showed a significant and substantial change in the (bio)chemical composition of individual cells resulting in virocells.

To challenge the results of the observed differences between cultures with and without phage addition, we applied the abovementioned multivariate analysis to two different time points of the same uninfected culture of *P. syringae*. This experiment was set out with the aim of testing the null hypothesis that the differences between

infected and noninfected cultures originates from variation during growth phases of cultures, which is known to exist in bacteria (9). The respective PCA (see Fig. S2 in the supplemental material), which also includes data from the infected cell culture, displays a difference of two time points along PC 2. However, the intragroup dissimilarity of the two individual time points was substantially lower than that for the population infected with phage, particularly along the major component of the PCA. Although the MRPP testing for differences between the uninfected cultures at the two time points resulted in a significant $P$ value (0.002), the chance-corrected within-group agreement was less than a sixth (0.009) of those identified for differences between cultures with and without phage addition. Moreover, comparing the combination of both time points of the uninfected culture to one with phage addition, we identified a highly significant difference (MRPP, $P < 0.001$, A = 0.06). Based on these observations the null hypothesis was rejected, supporting the working hypothesis that uninfected cultures can be distinguished from cultures with virocells using Raman spectroscopy.

**Differentiating wavenumbers of uninfected cells and virocells are attributable to nucleic acid and protein Raman shifts in *P. syringae*.** To investigate the exact differences between cultures with and without phage addition as displayed in the PCA (Fig. 2B), we used the system of *P. syringae-phi6* for an in-depth statistical analysis. Comparing the contrast plot of phage-infected and noninfected cultures with the major two components of the PCA highlighted the contribution of the individual wavenumbers that discriminate the two groups (cultures with and without phage addition) (Fig. 2A). Six wavenumbers were identified as local maxima/minima displaying the differences between the average spectra of the two groups with a high contribution to the PCA or a high density at the contrast plots, and these are assigned to their respective biomolecules in Table 1.

Three of the wavenumbers with the highest density in the contrast plot were assigned to nucleic acids (785, 1,483, and 1,576 1/cm), of which one was significantly higher in cultures infected with phage based on a Wilcoxon test ($P_{785} = 0.15$, $P_{1483} = 0.71$, $P_{1576} = 2.2 \cdot 10^{-16}$, respectively). In contrast, peaks assigned to proteins (1,003 and 1,671 1/cm) and lipids (1,448 1/cm) are more prominent in the control sample, and the corresponding $P$ values of the proteins were significant (Wilcoxon test, $P_{1003} = 1.7 \cdot 10^{-15}$,

**TABLE 1** Wavenumbers assigned to biomolecules of microbial cells and their density in the contrast plots of infected and uninfected cells

| Wavenumber (1/cm) | Density[a] of: | | Peak assignment (14, 23, 24, 39) |
| --- | --- | --- | --- |
| | *P. syringae* | *B. subtilis* | |
| 623 | −0.0103 | 0.0000 | 623 adenine |
| 645 | −0.0047 | 0.0003 | 645 cytosine, adenine |
| 669 | 0.0139 | 0.0066 | 668 guanine |
| 726 | 0.0031 | 0.0180 | 723/724 adenine |
| 748 | −0.0094 | −0.0154 | 740 thymine |
| 784 | 0.0634 | 0.03444 | 785 cytosine/uracil |
| 855 | −0.0399 | 0.0049 | 848 ribose/O-P-O stretch |
| 902 | −0.0471 | −0.0042 | Various metabolites |
| 961 | −0.0598 | 0.0006 | 960 Valine/leucine |
| **1,005** | **−0.1428** | **0.0020** | **1,004 Phenylalanine** |
| 1,034 | −0.0626 | 0.0014 | 1,035 proteins/1034 phosphoenolpyruvate |
| 1,096 | −0.0084 | 0.0078 | 1,101 PO$_2^-$ |
| 1,175 | −0.0207 | −0.0035 | 1,174 L-histidine |
| 1,241 | −0.0018 | 0.0189 | 1,230–1,310 amide III interval, 1240 uracil |
| 1,336 | 0.0442 | 0.0072 | 1,337 adenine |
| 1,452 | −0.0466 | −0.0090 | 1,440 lipids |
| 1,482 | 0.0922 | 0.0346 | 1,482 guanine/adenine |
| **1,577** | **0.1333** | **−0.0207** | **1,573 adenine, guanine** |
| **1,671** | **−0.1031** | **0.0397** | **1,640–1,680 amide 1, 1,671 thymine** |

[a]A high positive density refers to prominence in infected cells, and a negative value refers to wavenumbers more prominent in the control sample. Boldface indicates wavenumbers chosen for calculating the ratio for differentiating virocells from uninfected cells.

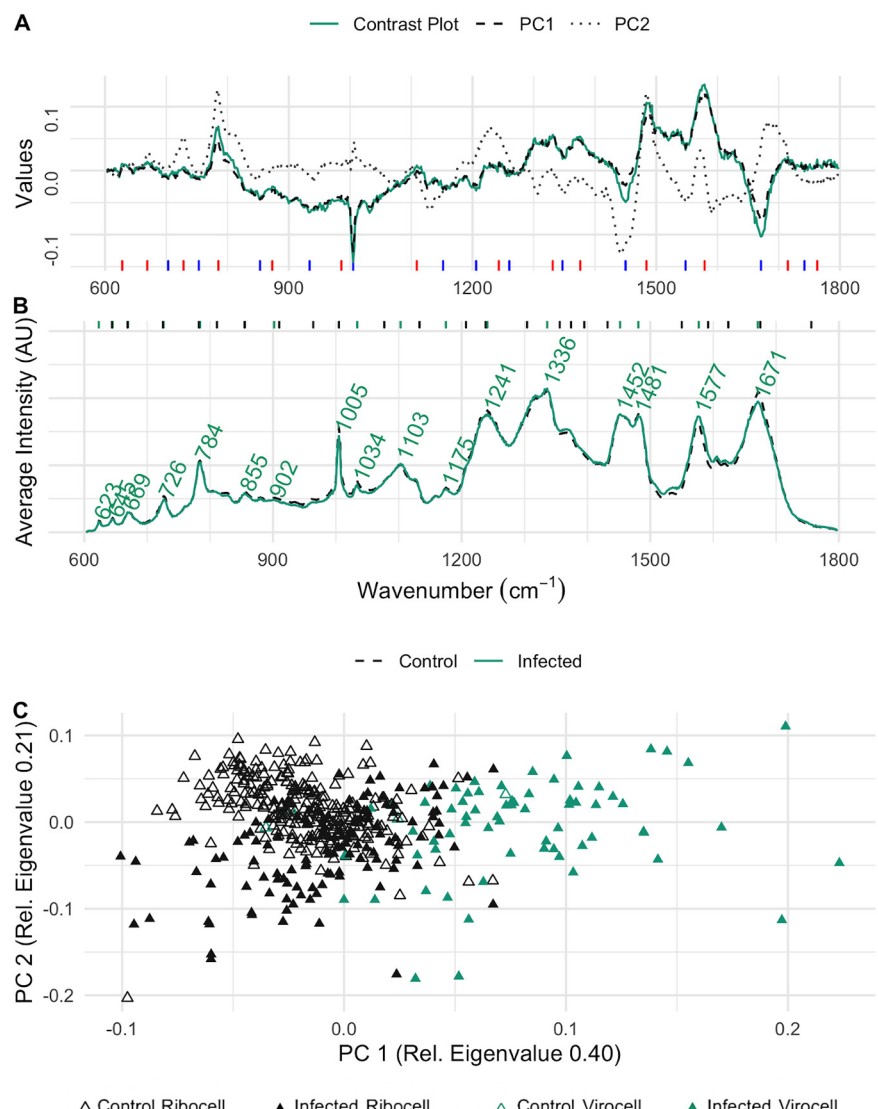

**FIG 3** Evaluation of wavenumbers for virocell identification in *P. syringae*. (A) Contrast plot (green) of potential infected cells compared to the wavenumber influence on PC 1 (black, dashed) and PC 2 (gray, line dotted). Blue lines at the bottom indicate wavenumbers that decreased in virocells, and red lines indicate wavenumbers increasing in virocells. (B) Average Raman spectra of the samples with (green, solid) and without (black, dashed) phage addition. Green lines at the top indicate the positions of the labeled peaks in the Raman spectra, and black lines indicate peak maxima of the variable importance on projection of the orthogonal partial least-square model. (C) PCA of single-cell Raman spectra from infected cultures (filled triangle) and cultures without phage addition (empty triangle); virocells identified based on the determined ratio are shown as filled triangles in green.

$P_{1671} = 2.2 \cdot 10^{-16}$, $P_{1448} = 0.71$, respectively). It is noteworthy that the wavenumbers associated with (highly) significant changes (1,003, 1,576, and 1,671 1/cm) contributed to PC 1, while the other three with insignificant changes between phage infected and noninfected contributed more to PC 2 (785, 1,440, and 1,483 1/cm) (Fig. 3A). The intensities (*I*) of three wavenumbers with significant *P* values were used to determine a differentiator for univariate differentiation of *P. syringae* virocells from uninfected cells (Equation 1):

$$\text{Ratio}_{\text{virocell}} = \frac{\text{Nucleic acids}}{\text{Proteins}} = \frac{I_{1576} \times 2}{I_{1003} \times I_{1671}} \tag{1}$$

with wavenumbers assigned to proteins (1,003 and 1,671 1/cm) in the denominator and the nucleic acid peak (1,576 1/cm) in the numerator. The Shapiro test demonstrated that the ratios based on spectra of the control sample based on Equation 1

were normally distributed ($P = 0.07$) and those of the infected sample were not ($P = 5.99 \cdot 10^{-9}$), which was expected since the latter represent a mixture of virocells and uninfected cells. The calculated confidence intervals indicate that *P. syringae* cells of the control group do not exceed a ratio above 1.06 (99% probability), while this threshold was indeed exceeded (with a probability of 45%, 66 of 198 cells) in the sample after phage addition. Consequently, Equation 1 can be used to identify potential virocells in cultures of *P. syringae* (Fig. 2C).

**Validation of selected wavenumbers for virocell identification of *P. syringae* via VIP of the OPLS analysis model indicates high influence by peak shoulders.** The average Raman spectra of the control sample and the infected sample show clear differences in the intensity of prominent biomolecule peaks chosen for the ratio determination (Fig. 2B). However, plotting the peak maxima of the variable importance on projection (VIP) of the orthogonal partial least square model (OPLS) together with the average Raman spectra revealed that the differences in virocells are sometimes not only represented by the maximum of the peak but also by its shoulders. This is a fine detail that is overseen by just visually inspecting the average spectra. The peaks can be assigned to their biomolecular origin (chemical bond) since their position does not change with a change in the molecular environment. However, the width of the peak is dependent on the composition of the molecule surrounding the polarized bond (20). Although the intensity change cannot be determined between two groups, this approach confirms the selected wavenumbers for the determined ratio for virocell identification in *P. syringae*.

**Applicability of virocell identification across three different species.** Based on the differentiating ratio determined for virocells and uninfected cells of *P. syringae* (Equation 1), we tested its applicability to other microbial species by repeating the analysis performed with *P. syringae*-phi6 for *B. subtilis*-phi29 and *M. mazei*-methanosarcina spherical virus (MetSV). We calculated the ratios (Equation 1) for cultures with and without virus addition, which showed a significant difference for *P. syringae* and *B. subtilis* ($P < 0.0001$) (Fig. 4). By contrast, only a trend was revealed for the *M. mazei*-MetSV system ($P < 0.0649$) (Fig. 4) without visible differences in PCA and contrast plot analysis (Fig. S3).

As a model system for Gram-positive bacteria, *Bacillus subtilis* was chosen as a representative, as it is a well-studied model organism with its dsDNA phage *phi29* of the order *Caudovirales*, which is among the smallest known dsDNA phages (21). For the *B. subtilis*-phi29 system, a group of potential virocells could be differentiated from the control sample along PC 2 (Fig. 5C), yet the contrast plot (Fig. 5A) shows a lower density range than the one for *P. syringae* (Fig. 3A). The highest values contributing to spectra of infected cells were associated with nucleic acids and proteins (Wilcoxon test, $P_{785} = 3.4 \cdot 10^{-11}$, $P_{1483} = 3.0 \cdot 10^{-12}$, $P_{1003} = 0.15$, and $P_{1671} = 9.5 \cdot 10^{-12}$), while peaks with the wavenumbers for hydrocarbons and nucleic acids were enriched in uninfected cells ($P_{1131} = 1.7 \cdot 10^{-8}$, $P_{1550} < 2.2 \cdot 10^{-16}$, and $P_{1589} < 2.2 \cdot 10^{-16}$). Importantly, these identified wavenumbers included the same wavenumbers that were determined for the ratio (Equation 1) for *P. syringae*. Although the associated signals of biomolecules were inverted compared to those for *P. syringae*, i.e., proteins were substantially higher in designated virocells and nucleic acids declined, the respective ratio (equation 1) can still be used to identify potential virocells of *B. subtilis* in Raman spectroscopy (Fig. 5).

As an archaeal system, we chose the anaerobic methane producer *Methanosarcina mazei* and its first identified and isolated virus, MetSV, which is classified as a dsDNA virus (22). The PCA of *M. mazei* is presented in Fig. S3. In contrast to the bacterial systems, no difference between the control and infected cultures was observed. For instance, the chance-corrected within-group agreement of these cultures was lower than the chance-corrected within-group agreement across uninfected *Pseudomonas* cultures (after 24 h), displaying little variance in the spectra of ribocells and virocells of *M. mazei* (A = 0.006872 for *M. mazei*, A = 0.009298 for *P. syringae*). During spectral acquisition of *M. mazei* cells, the raw spectra displayed a high background signal. This background signal was associated with autofluorescence of the methanogen, since it

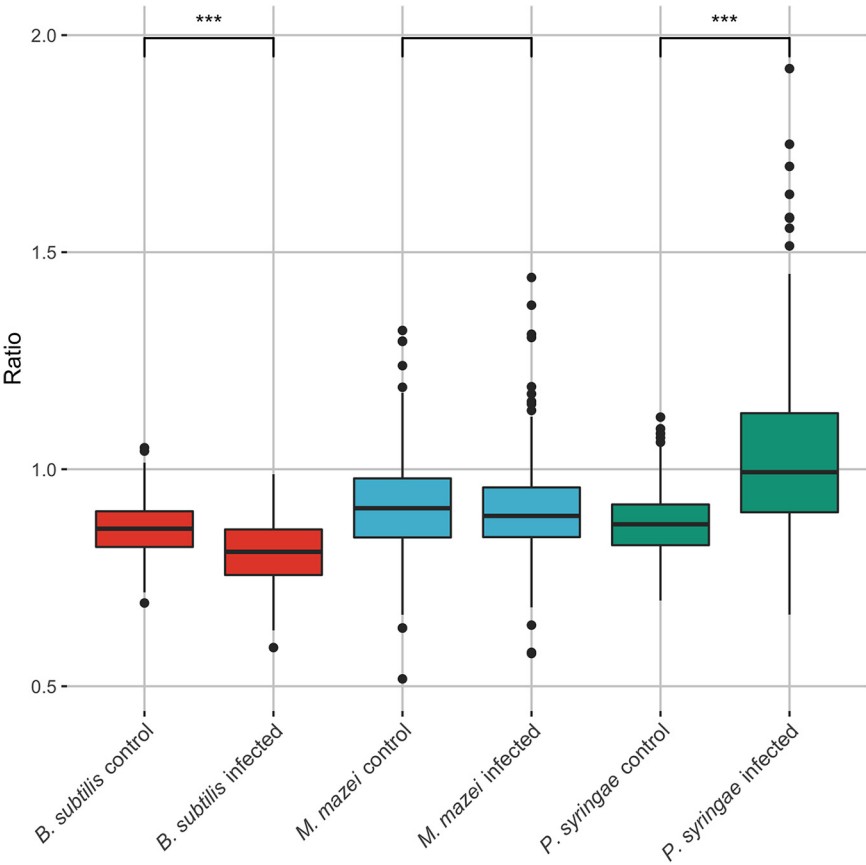

**FIG 4** Boxplot of the determined ratio for control (no virus addition) and infected (with virus addition) samples of *B. subtilis* (red), *M. mazei* (blue), and *P. syringae* (green). Asterisks indicate significance according to Wilcoxon test (***, highly significant, *P* < 0.0001; no asterisk, not significant, *P* < 0.06). For the exact number of spectra per sample see Table S1, and for a detailed multiple comparison across species (based on Dunn's test) see Table S2.

could be reduced by applying laser bleaching (30 s). This led to a higher dispersion of individual spectra of the same sample in ordination analysis. Figure S4 displays the dispersion of the archaeal sample set compared to the bacterial data sets. The bleaching time reduced the dispersion but was insufficient for enabling a differentiation between infected and noninfected samples.

## DISCUSSION

In this work, we measured several hundred individual bacterial and archaeal cells (see Fig. S4 in the supplemental material) to identify common changes in Raman spectra due to viral infections. One major challenge associated with measuring cultures of infected cells was their heterogeneity, meaning the culture consisting of uninfected cells and virocells at the same time. However, we were able to identify a specific ratio of Raman spectra that allowed us to differentiate virocells and ribocells in the cultures of *P. syringae* and *B. subtilis*. This ratio was based on the wavenumbers 1,003, 1,576, and 1,671 1/cm, which can be assigned to proteins and nucleic acid changes based on existing literature of recoded Raman spectra (14, 23).

**Overcoming challenges in identifying a Raman spectrum-based marker for virocells.** For identification of a Raman spectrum-based marker of virocells, it was mandatory to use univariate and multivariate statistics in concert. Neither univariate nor multivariate statistics alone were successful in identifying the respective wavenumbers necessary for the differentiation of virocells from uninfected cells.

To initially identify a set of wavenumbers that showed differences between these two cell types, we applied a multivariate analysis, resulting in six wavenumbers, which

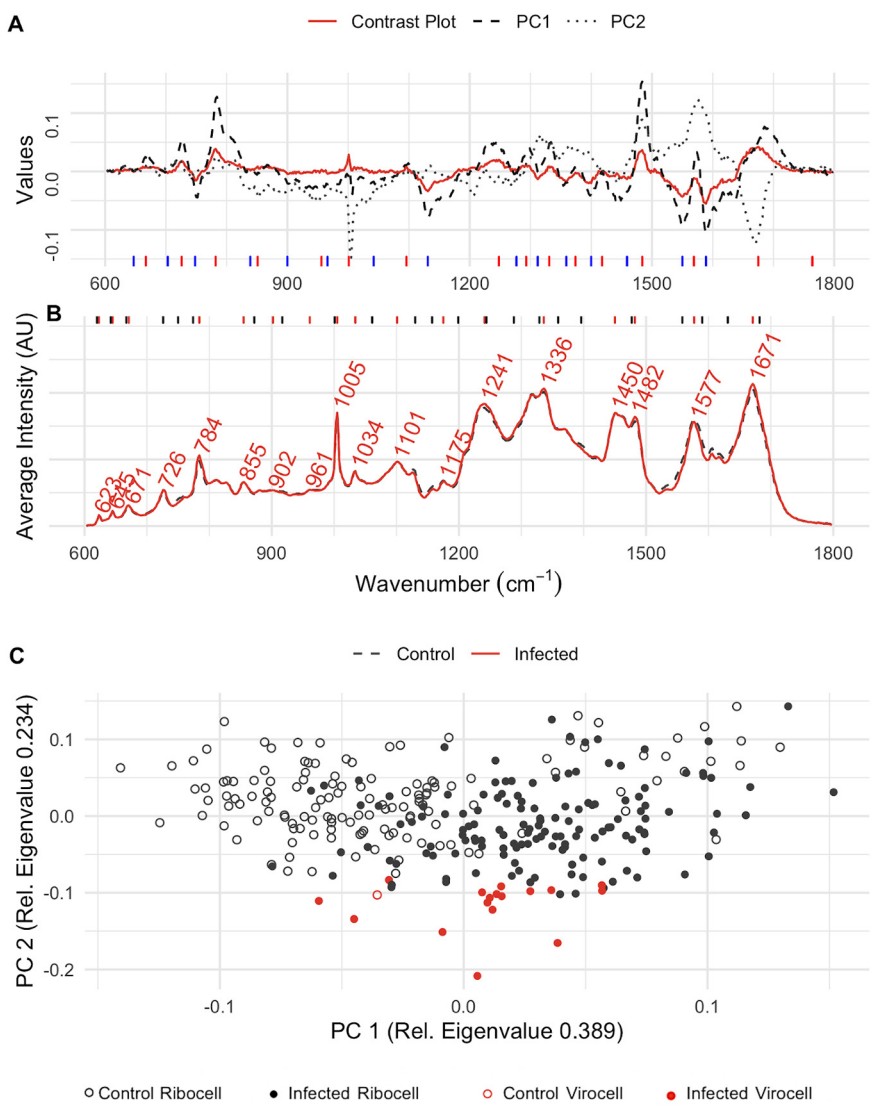

**FIG 5** Evaluation of wavenumbers for virocell identification in *B. subtilis*. (A) Contrast plot (red) of potential infected cells compared to the wavenumber influence on PC 1 (black, dashed) and PC 2 (gray, dotted). Long, blue lines at the bottom indicate wavenumbers that decrease in virocells, and red lines indicate wavenumbers increasing in virocells. (B) Average Raman spectra of the samples with (red, solid) and without phage (black, dashed) addition. Red lines at the top indicate the positions of the labeled peaks in the Raman spectra, and black lines indicate peaks of the OPLS importance. (C) PCA of single-cell Raman spectra from infected cultures (filled dots) and cultures without phage addition (empty dots); virocells identified based on the determined ratio are shown as filled dots in red.

were further filtered based on a Wilcoxon test to create the respective equation for differentiation of the two cell types. This was partly because multiple PCs can contribute to differences in statistical populations at various intensities, while we focused only on the two PCs with the greatest eigenvalues. Two peaks contributing substantially to PC 2 of both bacteria studied here are assigned to guanine (1,483 1/cm) (23) and the ring breathing of cytosine and uracil (785 1/cm) (14). Although this suggests a strong involvement of nucleic acid changes in uninfected versus virocells, the Wilcoxon test did not indicate a significant difference demonstrating an insufficient picture provided by multivariate data analysis (MRPP). On the other hand, using univariate statistics alone, the highest differences for the populations of *B. subtilis* did not occur at the maximum of the peak (1,579 1/cm), which we determined from using both methods.

mSystems®

Instead, the contrast plot had the highest values at the shoulders of the maximum peak, at 1,550 1/cm and 1,589 1/cm, suggesting that the peak position must be considered in Raman spectra via multivariate statistics. The reason for this phenomenon of the breadth of the peak can be traced back to the chemical environment of the molecule, as the Raman shift is characteristic for the polarized chemical bond (20, 24, 25). Several studies about differences of Raman spectra of packed and unpacked viral DNA/RNA and protein/oligonucleotide interactions have been performed in the past and describe altered base environments as the reason for the observation of such perturbations (20, 24, 25).

**phi29 likely causes a stress response in B. subtilis.** The determined equation for differentiating virocells from ribocells in the *P. syringae-phi6* system could also be applied to the *B. subtilis-phi29* system. However, the ratio used for the differentiation was significantly lower in the *B. subtilis* system, which is in stark contrast to the significantly higher ratio for *P. syringae*. The respective wavenumbers attributable to proteins (1,003 and 1,671 1/cm) showed an increase in intensity in *B. subtilis*, and nucleic acids (1,576 1/cm) appeared to decrease substantially during infection with *phi29*. A drop in nucleic acid content and increase in protein content (as observed here for virocells of *B. subtilis*) is complementary to multiple biological processes that can be observed for bacteria. Chemicals like ethanol can cause a similar change in the protein and nucleic acid content, which represents a stress response by the bacterium. This stress response was detected based on the same changes in the wavenumbers as those observed here (26). However, the induction of temperate phage in *B. subtilis* was shown to result in a decrease of the Raman shifts at 782 and 1,095 1/cm and only a slight decrease at 1,452 and 1,659 1/cm (27). The authors of the aforementioned study concluded that these measurements likely stem from the fact that the measured cell had ruptured and an empty cellular hull had been measured (consisting of proteins and lipids, while nucleic acids are lost during lysis). They used Raman shifts around 1,095 1/cm and 785 1/cm to measure the respective differences in the nucleic acid, neither of which showed a significant difference in our data sets. Comparing these previous findings to our results for *B. subtilis*, some likely cannot differentiate between *B. subtilis* cells showing a stress response and a respective virocell. We conclude that *phi29* causes a stress response in *B. subtilis* during infection, which we measured during Raman spectra acquisition.

**High sensitivity of Raman spectra mirrors different types of phage infection.** The changes in nucleic acid and protein content are contradictory in the *P. syringae* and the *B. subtilis* system and could not be attributed solely to complex stress responses but rather to different types of phages. While *phi6* infecting *P. syringae* is a nontailed RNA phage with a lipid membrane (15, 28), *phi29* is a DNA phage with a complex polypeptide structure consisting of a phage head and a phage tail (29). Consequently, an increase in the protein content during *phi29* replication can be associated with an increase in protein content in the cell. The wavenumber 1,671 1/cm was previously associated not only with the amides but also with thymine, a central component of DNA but not RNA (Table 1). Comparing the RNA phage *phi6* and the DNA phage *phi29*, we did observe a difference at the thymine concentration at this wavelength. A similar trend (increase in thymine/protein concentration) was also observed for the *M. mazei* system, which is also based on a DNA virus. We conclude that the putative increase of proteins measured at 1,671 1/cm stems from an increase in protein and thymine concentration at the same time, reflecting the difference in DNA and RNA phage used in the experiments.

Beyond the different types of phages, the relatively slow maturation of the *phi6* virions usually encompasses two different stages within the *P. syringae* host. After 45 min, 50-nm particles can be observed within the host, and after 80 min these particles are covered by the viral membrane (17). The plot of the PCA in Fig. 1 shows that infected cultures of *P. syringae* differed along both components. Component one was used for the ratio determination, but the ratio did not include spectra of individual virocells that showed a difference along component two. The shift of these virocells along PC2 was associated with a single wavenumber at 1,448 1/cm. This wavenumber indicates an

increase of lipids, which agrees with the production of lipid membranes for viral particle maturation (17). Consequently, we succeeded not only in identification of virocells of *P. syringae* but also in distinguishing the two infection stages during *phi6* maturation based on our Raman spectra.

**Conclusions.** Our data, encompassing 1,287 Raman spectra acquired for individual cells of three different microbial species with and without virus addition, suggest that at least bacterial virocells can be differentiated from uninfected cells. We present a ratio of three wavenumbers that can be utilized to quickly perform this differentiation, although the type of phage (RNA versus DNA) and different infection stages can influence the detection. Beyond detection, Raman spectra of individual cells are sensitive enough to capture essential information on the biology of individual phage-host systems. Namely, DNA and RNA phages and stress responses to the differentiation of maturation stages of phages within the microbial host cell can be robustly identified. We predict that the identification of such cells in batch culture experiments and ultimately in environmental samples will aid studying the biology of individual virocells and, thus, expand our understanding of the complex interplay of phages and hosts along with their associated biochemistry.

## MATERIALS AND METHODS

**Cultivation of model systems and sampling strategy.** Two cultures of *Pseudomonas syringae* (DSM21482) were incubated at 25°C with 150 rpm in tryptone soya broth (DSM medium 545). After 24 h, the cultures reached the exponential growth phase; 1 vol% glycerol stock of the phage *phi6* (DSM21518) was added to one culture, and the second culture was kept uninfected as a negative control. Samples for Raman microspectroscopy were taken prior to phage addition and 10 h after infection, indicated by a drop of the optical density (OD).

*Bacillus subtilis* (DSM5547) was incubated at 37°C with 150 rpm in DSM medium 545. After 4 h, the cultures reached the exponential growth phase, and 10 vol% of a phage *phi29* solution (DSM5546) was added to one culture; the second culture was kept uninfected as a control. The shaking was reduced to 80 rpm. Samples for Raman microspectroscopy were taken when the optical density dropped 2 h after infection (see Fig. S5 in the supplemental material).

*Methanosarcina mazei* (DSM3647) was grown in minimal medium under anaerobic conditions with an $N_2$-$CO_2$ (80:20) atmosphere in closed serum bottles without shaking at 37°C. As energy and carbon sources, 150 mM methanol and 40 mM acetate were added. Furthermore, medium was supplemented with 2 mM cysteine and 1 mM sodium sulfide as described previously (30, 31). When the sample reached turbidity at an OD at 600 nm ($OD_{600}$) of approximately 0.2, cultures were infected with 1% filtrated MetSV lysate (22). Samples for Raman spectroscopy were taken anaerobically before and 180 min and 210 min after infection.

**Sample preparation for Raman microspectroscopy.** Samples for Raman microspectroscopy were taken at respective time points from the model systems (described above); 1 ml of the culture was washed with 1 ml 1× phosphate-buffered saline (PBS; pH 7.4; Sigma-Aldrich), followed by resuspension in 0.45 ml 1× PBS and 0.15 ml 4% formaldehyde (Thermo Scientific) solution (fixation at 4°C for 3 h). Afterwards the sample was again washed in 0.5 ml 1× PBS and dehydrated at room temperature in 50 vol% and 80 vol% ethanol (Fisher Scientific) for 10 min each. Finally, the preparation was stored in 0.15 ml 96% ethanol at −20°C until spectral acquisition. Throughout all steps mentioned above, washing was done by pelleting of samples via centrifugation at 2,000 × *g* for 10 min, followed by discarding the supernatant.

**Raman spectral acquisition.** Raman spectral acquisition was performed using a Renishaw in via a Raman microspectroscope with a 532-nm Nd:YAG laser and 1,800-l/mm grating equipped with a Leica DM2700M microscope. A 100× dry objective with a numerical aperture of 0.85 was used. Daily calibration was performed using a silicon wafer (Renishaw). For each dehydrated sample (preparation as described above), a drop was placed on a highly polished steel slide (Renishaw) and air dried. Figure S6 displays the even distribution of cells on the slide. For *Pseudomonas syringae*, a spectral acquisition of 25 to 30 s at 10% laser power was used, and for *Bacillus subtilis*, three accumulations of 25 s and 5% laser power were used. For cells of *Methanosarcina mazei*, a 15-s bleaching step prior to 30-s measurement at 5% laser power was necessary to reduce the florescent background. At least 50 cells per drop were measured, and at minimum three drops per sample were used.

**Multivariate statistical analyses.** The spectra were imported to R (32) as SPC files and processed using the R package *MicroRaman* (12). The spectral data were trimmed to a range of 600 to 1,800 1/cm. After background subtraction using the statistics-sensitive nonlinear iterative peak-clipping (SNIP) algorithm (33), data were normalized using total ion current (TIC) (34). These preprocessed data were used to calculate principal component analyses (PCA) (35) and dendrograms based on Euclidian distance (Ward D2 clustering) (32). PCA results were compared to principal coordinate analyses (36) based on spectral contrast angle dissimilarities (12). Spectra of cells burnt during spectral acquisition, spectra of low intensity, and those containing cosmic rays were identified and removed from the data set. Wavenumbers causing differences between infected and uninfected spectra were identified using a contrast plot (12) and the influence on

the principal components. Differences between the samples were assessed via a multiresponse permutation procedure (MRPP) using 999 Monte Carlo permutations (18, 19).

An orthogonal partial least square analysis (OPLS) (37) was performed on the baseline-corrected data. The spectra were divided into "Species_control" or "Species_infected" according to the sample their originated from. The variable importance on projection (VIP) for each wavenumber in the range of 600 to 1,800 1/cm was determined and compared with the density of the contrast plot and the principal components.

The mean spectrum of each class was calculated by determining the mean intensity at each wavenumber.

**Determination of differentiating ratio of virocells and uninfected cells.** Different combinations of the intensities of the three wavenumbers with the most influence in the contrast plots (contrasting virocells and uninfected cells) of the *P. syringae-phi6* system were further analyzed. The average intensities and the standard deviations were calculated for the normalized data of the uninfected cells and potential virocells. A Shapiro test for normal distribution then was performed, and a Wilcoxon test for nonnormally distributed data was used to test if the data from infected and uninfected cells show a significant difference. For each ratio, a cutoff value was defined to declare a cell was infected. The 99% confidence interval was calculated for the infected group and the control group, and afterwards the number of false-positive spectra inside the control group was determined. The results derived from the *P. syringae-phi6* system for identification of differentiating wavenumbers was then applied to the other virus-host systems and a Dunn's test was performed to differentiate between host type coupled to infected/ uninfected cultures (https://github.com/cran/dunn.test) (38).

To identify the respective Raman spectra and relate them to biomolecules, we followed various publications by G. J. Thomas and coworkers, which resulted in a collection of Raman spectra of nucleic acids and proteins (14), and De Gelder et al. (23), who conducted a study on pure solutions of biomolecules. The assignments are summarized in Table 1.

## SUPPLEMENTAL MATERIAL

Supplemental material is available online only.
**FIG S1**, TIF file, 15.6 MB.
**FIG S2**, TIF file, 7.8 MB.
**FIG S3**, TIF file, 15.6 MB.
**FIG S4**, TIF file, 7.8 MB.
**FIG S5**, TIF file, 6.7 MB.
**FIG S6**, JPG file, 0.1 MB.
**TABLE S1**, PDF file, 0.03 MB.
**TABLE S2**, PDF file, 0.1 MB.
**TABLE S3**, PDF file, 0.02 MB.

## ACKNOWLEDGMENTS

This work was funded by German Research Foundation (DFG grant PR1603/2-1). I.M. was supported by the German Academic Scholarship Foundation, and A.J.P. acknowledges funding by the Ministry of Culture and Science of North Rhine-Westphalia (Nachwuchsgruppe "Dr. Alexander Probst"). We acknowledge support by the Open Access Publication Fund of the University of Duisburg-Essen.

We thank Sabrina Eisfeld and Agathe Materla for lab management and technical assistance. Priyanka Mishra and Rainer Meckenstock are acknowledged for support with the Raman microspectroscope.

We declare no conflict of interest.

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
