## [Reviewer comments · mSystems]

Label-free Raman microspectroscopy for identifying prokaryotic virocells

Indra Monsees, Victoria Turzynski, Sarah Esser, André Soares, Lara Timmermann, Katrin Weidenbach, Jarno Banas, Michael Kloster, Bánk Beszteri, Ruth Schmitz, and Alexander Probst

Corresponding Author(s): Alexander Probst, University of Duisburg-Essen

Review Timeline:

Submission Date:

January 11, 2022

Accepted:

January 11, 2022

Editor: Joanne Emerson

Reviewer(s): The reviewers have opted to remain anonymous.

Transaction Report:

DOI: <https://doi.org/10.1128/msystems.01505-21>

January 11, 2022

Dr. Alexander J Probst
University of Duisburg-Essen
Campus Essen - Biofilm Centre
Universitätsstr. 5
Essen 45141
Germany

Re: mSystems01505-21 (Label-free Raman microspectroscopy for identifying prokaryotic virocells)

Dear Dr. Alexander J Probst:

Your manuscript has been accepted, and I am forwarding it to the ASM Journals Department for publication. For your reference, ASM Journals' address is given below. Before it can be scheduled for publication, your manuscript will be checked by the mSystems senior production editor, Ellie Ghatineh, to make sure that all elements meet the technical requirements for publication. She will contact you if anything needs to be revised before copyediting and production can begin. Otherwise, you will be notified when your proofs are ready to be viewed.

Publication Fees:

We recognize that the video files can become quite large, and so to avoid quality loss ASM suggests sending the video file via <https://www.wetransfer.com/>. When you have a final version of the video and the still ready to share, please send it to mSystems staff at mssystemsjournal@msubmit.net.

For mSystems research articles, if you would like to submit an image for consideration as the Featured Image for an issue, please contact mSystems staff at mssystemsjournal@msubmit.net.

Sincerely,

Joanne Emerson
Editor, mSystems

Journals Department
Table S2: Accept
Fig. S4: Accept
Table S1: Accept
Fig. S2: Accept
Fig. S3: Accept
Table S3: Accept
Fig. S6: Accept
Fig. S1: Accept
Fig. S5: Accept